# VERSATILE MOTION-LANGUAGE MODELS FOR MULTI-TURN INTERACTIVE AGENTS

## ABSTRACT

Recent advancements in large language models (LLMs) have greatly enhanced their ability to generate natural and contextually relevant text, making AI interactions more human-like. However, generating and understanding interactive human-like motion, where two individuals engage in coordinated movements, remains a challenge due to the complexity of modeling these coordinated interactions. Furthermore, a versatile model is required to handle diverse interactive scenarios, such as chat systems that follow user instructions or adapt to their assigned role while adjusting interaction dynamics. To tackle this problem, we introduce **VIM**, short for the Versatile Interactive Motion language model, which integrates both language and motion modalities to effectively understand, generate, and control interactive motions in multi-turn conversational contexts. To address the scarcity of multi-turn interactive motion data, we introduce a synthetic dataset called **INTER-MT$^2$**; where we utilize pre-trained models to create diverse instructional datasets with interactive motion. Our approach first trains a motion tokenizer that encodes interactive motions into residual discrete tokens. In the pre-training stage, the model learns to align motion and text representations with these discrete tokens. During the instruction fine-tuning stage, VIM adapts to multi-turn conversations using INTER-MT$^2$. We evaluate the versatility of our method across motion-related tasks—motion-to-text, text-to-motion, reaction generation, motion editing, and reasoning about motion sequences. The results highlight VIM's versatility and effectiveness in handling complex interactive motion synthesis.

## 1 INTRODUCTION

Agents that reflect how humans communicate and interact with each other through motion have the potential to revolutionize our interaction with technology. By capturing the subtleties of human communication, including gestures, expressions, and interactive behaviors, these agents can offer more intuitive and natural interfaces. This holistic understanding enables technology to adjust its responses and behaviors based on the user's physical motions and situational context, leading to more personalized and engaging interactions. Such capabilities are crucial for enhancing support across various domains, including robotics, virtual humans, entertainment, and more.

Recent advancements in large language models (LLMs) (Dubey et al., 2024; Team et al., 2024; Yang et al., 2024) have demonstrated significant potential in generating human-like text and understanding complex linguistic interactions. They have even extended their capability to multi-modal contexts, successfully integrating various input sources such as images, speech, and videos (Ge et al., 2024; Liu et al., 2024; Chen et al., 2024b; Tang et al., 2024; Shu et al., 2023). Building upon these developments, there is a growing interest in incorporating human (or robot) motion as a new modality (Jiang et al., 2024; Chen et al., 2024a), leading to the emergence of the "motion-language models" (MLM). However, existing approaches (Zhang et al., 2023; Guo et al., 2024a; 2022; Zhang et al., 2024d; Cai et al., 2024) often focus on single tasks, such as text-to-motion or motion-to-text translation, and consider only single motions without interactions. This limitation hinders the agents' ability to handle scenarios involving multi-agents, complex interactions, and multi-turn conversations.

Beyond modeling the motion of a single person, interactive motion, where two individuals participate in interactions, allows the model to learn about social behavior. Modeling such interactions requires versatility to effectively control interactions, allowing users to provide instructions, assign

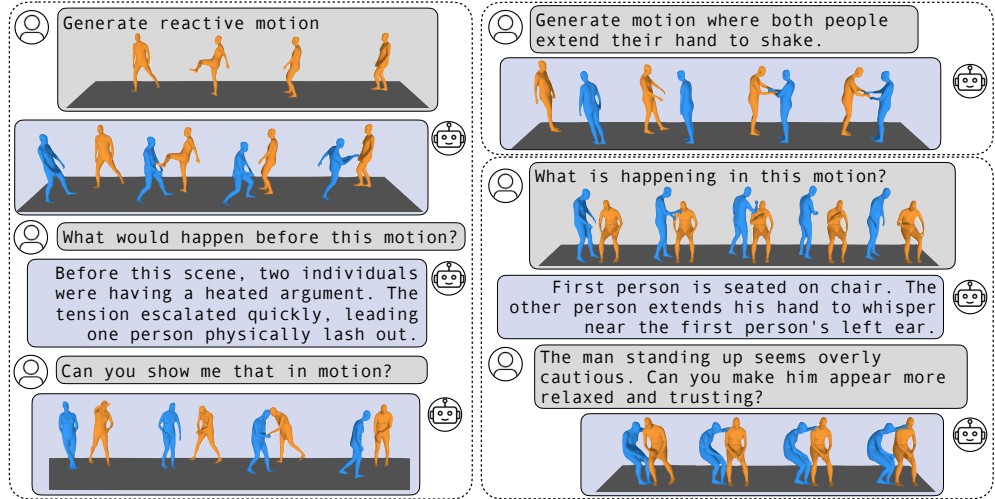

Figure 1: We introduce VIM, a versatile interactive motion language model. Left: reaction generation, motion reasoning, and generation. Right top: text-to-motion generation. Right bottom: Motion editing and motion understanding.

roles, or modify behaviors. In this paper, we aim to build a unified yet versatile motion-language model designed to generate, control, and comprehend sophisticated interactive motions.

One of the primary challenges in developing these interactive agents is the lack of multi-turn interactive motion data. Datasets containing motions of two individuals interacting with each other, along with multi-turn conversational instructions, are scarce and challenging to collect. This makes it difficult for models to learn the nuances of interactive motions and multi-turn dynamics.

To address this, we present a new synthesized dataset called **INTER-MT$^2$**, which includes various instructions about the interactive motions, in a multi-turn conversational format. We utilize a large language model to produce diverse instructions with motion captions and a diffusion-based text-to-motion model to generate corresponding motions. We expect that leveraging such foundation models to construct training data allows the model to generalize more effectively with prior knowledge.

Building upon our synthesized dataset, we present **VIM**, a Versatile Interactive Motion-language model designed for multi-turn conversations involving interactive motions. We pursue the versatility of VIM through a unified architecture that can simultaneously input and output both motion and text modalities. Based on the pre-trained LLMs, our training process can be divided into three stages: (1) training of the interaction motion tokenizer, (2) pre-training for motion and text representation alignment, and (3) instruction tuning with our synthesized dataset, INTER-MT$^2$, to handle more complex and multi-turn instructions. This enables VIM to effectively comprehend, generate, and control interactive motions, as illustrated in Figure 1. To assess VIM's capabilities, we introduce new evaluation protocols that evaluate its performance across various motion-related tasks. This include editing motions and reasoning about motion sequences based on contextual cues, highlighting its versatility and effectiveness in complex motion interaction scenarios.

In summary, the main contributions of this paper are threefold: (1) We propose VIM that can simultaneously process and generate both motion and text modalities, along with a three-stage training pipeline consisting of motion tokenizer training, pre-training for modality alignment, and instruction tuning. (2) We present INTER-MT$^2$, a multi-turn interactive motion-text dataset, to address the lack of multi-turn interactive motion data. (3) We introduced a new evaluation protocol to evaluate the performance of motion-language models on complex motion interaction scenarios.

## 2 RELATED WORK

**Human Motion Modeling & Control**    Advancements in human motion modeling have driven significant progress in both motion generation and control. Diffusion-based methods, such as MDM

(Tevet et al., 2023), FG-T2M (Wang et al., 2023), and MotionDiffuse (Zhang et al., 2024a) excel at synthesizing realistic human motions from the text. Transformer-based models with vector quantization, such as TM2T (Guo et al., 2022) and T2M-GPT (Zhang et al., 2023), effectively capture complex motion patterns. MoMASK (Guo et al., 2024a) improves motion granularity with residual tokenizers. For motion editing, some approaches focus on style transfer (Aberman et al., 2020; Guo et al., 2024b) or specific body part modifications (Zhang et al., 2024a; Kim et al., 2023). FineMoGEN (Zhang et al., 2024c) offers fine-grained motion synthesis based on user instructions. MEOs (Goel et al. (2024)) use captions and large language models to identify frames and body parts to edit, while MotionFix (Athanasiou et al. (2024)) conditions diffusion models on both source motion and edit text for seamless motion edits. However, these models usually target single tasks (*e.g.*, text-to-motion, or motion editing) and lack versatility in handling input and output of both motion and text simultaneously in a unified architecture.

**Motion Language Model**   Recent developments in motion language models have aimed to achieve versatility across various motion-related tasks. MotionGPT (Jiang et al., 2023) demonstrates versatility in motion comprehension and generation based on a unified framework. MotionChain (Jiang et al., 2024) introduces a multi-turn conversational system for interpreting and generating motions within dialogue contexts, including image inputs. Zhou et al. (2024) introduces AvatarGPT integrating motion generation and planning ability in motion large language model. Some studies, like Chen et al. (2024a), expand modalities into speech, music, and videos but focus primarily on comprehension rather than generation. Zhang et al. (2024b) proposed unified models for generating motion from various input modalities. $M^3$-GPT, from Luo et al. (2024), models speech, music, text, and motion interchangeably. However, modeling *interactive motions* in versatile large models remains under-explored. While some efforts, such as Wu et al. (2024), address this direction, they often lack multi-turn interactions and complex reasoning abilities. Our work addresses this gap with a model trained on our synthesized INTER-MT$^2$ dataset, enabling the understanding and generation of interactive motions in multi-turn conversations with advanced reasoning capabilities. This approach facilitates more nuanced, context-aware motion generation in complex interactive behaviors.

**Human-Human Interactive Motion Modeling**   Modeling human-human interactions has garnered increasing attention in recent research. Several multi-person interaction datasets (Ng et al., 2020; Fieraru et al., 2020; Yin et al., 2023) have been developed, and recent efforts like Inter-X (Xu et al., 2024a) and InterHuman (Liang et al., 2024) have collected interactive motions paired with textual descriptions for text-based motion control. In text-to-motion tasks, InterGEN (Xu et al., 2024a) introduces a diffusion-based model with spatial constraint loss. PriorMDM (Shafir et al., 2024) leverages pre-trained motion diffusion models with slim communication blocks. For reaction generation, ReMoS Ghosh et al. (2023) synthesizes reactive motion using spatio-temporal cross-attention, while ReGenNet Xu et al. (2024b) employs a transformer-based model with distance-based interaction loss to predict human reactions. While existing models have advanced interactive motion modeling, they lack versatility and focus on specific tasks, failing to capture complex multi-turn dynamics. To address this, we introduce INTER-MT$^2$, enabling agents to generate sophisticated motions, respond to instructions, adapt roles, and adjust behaviors based on context.

## 3   INTER-MT$^2$: INTERACTIVE MULTI-TURN MOTION-TEXT DATASET

Current datasets (Yin et al., 2023; Liang et al., 2024; Xu et al., 2024a) for modeling interactive motions lack sufficient diversity in instructions and do not include multi-turn conversations. To address this gap, we introduce **INTER-MT$^2$**: **INTER**active **MUTI**-Turn **M**otion-**T**ext dataset. This dataset covers a variety of interactive motion scenarios with multi-turn conversations, diverse instructions, and spatiotemporally aligned motions between two individuals.

Collecting diverse instructional data for interactive motions and multi-turn conversation samples poses significant challenges. The first challenge is obtaining instruction and conversational data that encompass complex reasoning and generation capabilities. Leveraging pre-trained foundation models to generate a broad range of instructions can enrich the dataset with diverse and intricate examples. The second challenge is acquiring motion data that aligns with these text instructions. Functional approaches using rule-based methods may struggle to maintain spatial and temporal constraints in complex interactive scenarios, while retrieval-based methods are limited by dependence

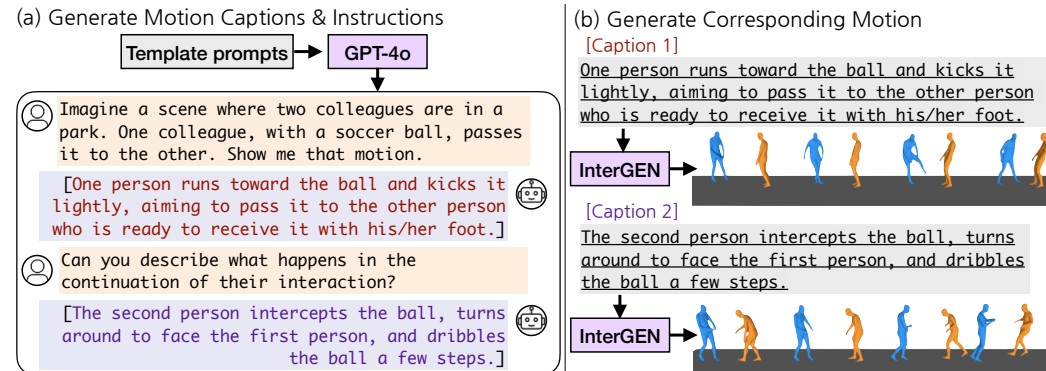

Figure 2: Overview of synthetic data generation for multi-turn conversations with interactive motions. (a) Motion captions and instructions are generated using GPT-4 based on interactions between two characters, followed by (b) the corresponding motion being synthesized using the InterGEN.

on existing cases and lack diversity. Alternatively, generative approaches using pre-trained models show promise in producing diverse, complex text-to-motion sequences, offering more flexibility for modeling interactive motions.

We utilize the Inter-X (Xu et al., 2024a) and InterHuman (Liang et al., 2024) datasets as the foundation for building our datasets. We further employ GPT-4o (OpenAI, 2024) to generate motion captions and conversational instructions for a variety of tasks, such as motion editing, reasoning, and story generation, enhancing the model's versatility. Motion captions are sourced from these base datasets or generated by large language models (LLMs). We utilize the state-of-the-art text-to-motion diffusion model, InterGEN Liang et al. (2024), to generate corresponding motions that align with the generated caption from LLMs. Our data collection pipeline, shown in Figure 2, comprises 82K samples of multi-turn conversational data involving interactive motions, including 96K of synthesized interactive motions and 56K motions from the source dataset.

## 4 VIM: VERSATILE INTERACTIVE MOTION-LANGUAGE MODEL

In this section, we introduce VIM, a versatile interactive motion language model, designed to incorporate multi-turn conversations considering both language and interactive motion as input or output modality. First, we will explain the underlying philosophy behind our design choices for the model architectures, followed by a detailed description of the training methodologies. Then, we introduce advanced interactive motion tasks in multi-turn conversations.

### 4.1 NOTATIONS

Formally, we denote interactive motion from two individual $a$ and $b$ as $\{\mathbf{m}_a, \mathbf{m}_b\}$, following non-canonical representation from Liang et al. (2024) based on SMPL-X structure (Pavlakos et al., 2019). Each timestep of the motion $\mathbf{m}^i = [\mathbf{j}_g^p, \mathbf{j}_g^v, \mathbf{j}^r, \mathbf{c}^f]$ is composed of global joint positions $\mathbf{j}_g^p \in \mathbb{R}^{3N_j}$, global joint velocities $\mathbf{j}_g^v \in \mathbb{R}^{3N_j}$, 6D representation of local rotations $\mathbf{j}^r \in \mathbb{R}^{6N_j}$, with the number of joints $N_j$, and binary ground contact features $\mathbf{c}^f \in \mathbb{R}^4$. We aim to train a motion language model $p_\theta$ which can model texts and motions for both inputs and outputs. We define input as an instruction or previous context and output as an answer, with the template below

$$\text{User}: X_u^1 \texttt{<Instruction>} \quad \text{Assistant}: X_a^1 \texttt{<Answer>}$$
$$\text{User}: X_u^2 \texttt{<Instruction>} \quad \text{Assistant}: X_a^2 \texttt{<Answer>} \quad \cdots$$

where $X_u$ and $X_a$ are composed of both a mixture of text modalities and motion modalities.

### 4.2 ARCHITECTURE

Our architecture for modeling and generating interactive motions consists of three primary components: encoders (tokenizers), a large language model block, and decoders. This design allows

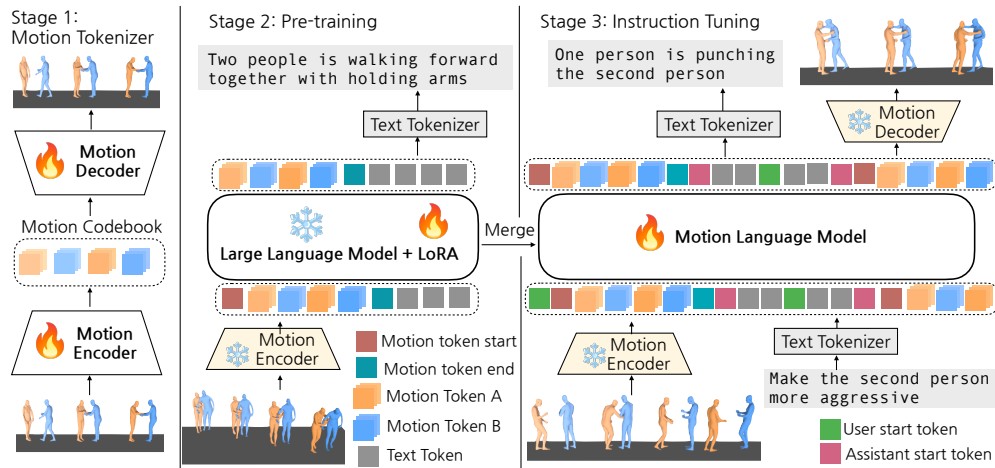

Figure 3: Method Overview. Stage 1 involves training a motion tokenizer that encodes and decodes interactive motion data. In Stage 2, we pre-train the model by integrating motion and text data, allowing it to learn the alignment between text and motion. Stage 3 focuses on Instruction Tuning, fine-tuning the model to follow instructions and improve its responsiveness to conversational cues.

for the integration of both motion and text data within a unified framework. For motion, we use a residual vector quantized variational auto-encoder (RQ-VAE (Lee et al., 2022; Guo et al., 2024a)) as a tokenizer. Vector quantized variational auto-encoders (Van Den Oord et al., 2017) are effective, but their quantization causes information loss and reduces reconstruction quality, which is critical for accurately modeling interactive motions. The motion encoder $\mathcal{E}_M$ applies 2D convolutions to motion features along the time axis, converting motion pairs $\mathbf{m}_a, \mathbf{m}_b$ into latent vectors $\{\mathbf{z}_a^{1:L}, \mathbf{z}_b^{1:L}\} = \mathcal{E}_M(\{\mathbf{m}_a^{1:M}, \mathbf{m}_b^{1:M}\})$, where $M$ is a motion length and $L = M/l$ with down-sample rate $l$. Then the latent vectors $\mathbf{z}$ are quantized by RQ-VAE as an ordered $D$ discrete codes:

$$\mathcal{RQ}(\mathbf{z}^i; \mathcal{C}, D) = (k_1^i, \cdots, k_D^i) \in [K]^D \tag{1}$$

where $\mathcal{C}$ is the codebook, $K = |C|$, and $k_d^i$ is code of $\mathbf{z}$ at timestep $i$ and depth $d$. These discrete codes form a motion vocabulary. For text, we use a standard text tokenizer to process textual instructions and descriptions into tokens.

Tokens are then proceeded to the language model block, which serves as the central processing unit. In this work, we have utilized the LLaMA-3.1-8B (Dubey et al., 2024) model as a base model. We integrate motion tokens with text tokens using a unified vocabulary, which combines both the text and motion vocabularies into one, with special tokens added to mark the start and end of the motion sequences. This shared token space enables the model to efficiently process and generate both modalities for motion-related tasks. Interactive motion is represented as $X_m = \{k_{1:D}^{1;a}, k_{1:D}^{1;b}, \cdots, k_{1:D}^{L;a}, k_{1:D}^{L;b}\}$, where $X_m$ is a sequence of motion represented in unified vocabulary and $k_{1:D}^{i;a} \in [K]^D$ is the $i$-th token of motion $a$.

Finally, the decoding stage reverses the encoding process. For motion, the decoder $\mathcal{D}_M$ projects the quantized features $\hat{\mathbf{z}}^i = \sum_{d=1}^D \mathbf{e}(k_d^i)$, where $\mathbf{e}$ is codebook embedding, back into motion sequences using 1D convolution. Text decoding follows standard language model decoding procedures.

### 4.3 TRAINING

**Motion Tokenizer** The first stage is to train a motion tokenizer composed of an encoder, decoder, and quantizer. We followed the original objective functions from Lee et al. (2022) to train this model, minimizing the reconstruction loss, the codebook loss to align the encoder's outputs with the codebook, and the commitment loss to ensure encoder consistency. Once the encoder and decoder are optimized, we maintain this model frozen during the rest of the training stage.

**Pre-training Strategy** In the pre-training stage, we train a pre-trained large language model to align motion representations with textual representations. We design tasks including motion-to-

text, text-to-motion, motion prediction, and reaction generation to train the model, leveraging paired datasets like Inter-X Xu et al. (2024a) and InterHuman Liang et al. (2024). Using the template from Jiang et al. (2023), we create input sequences $y$ from motion sequences $X_m$ and the corresponding motion caption. Since both motion tokens and text tokens are discrete, we train our model with the general language modeling next-token prediction objective: $\mathcal{L} = -\log \sum^T p_\theta(y_i|y_{<i})$, where $T$ is the length of the multi-modal sequence and $i$ only counts when the text token appears at position $i$. To improve training efficiency, we train the LLM using a low-rank adaptor (LoRA) (Hu et al., 2022), similar to Ge et al. (2024). We then merged the LoRA parameters to the LLM backbone for further training. Furthermore, due to a limited number of interactive motion data, we also leverage larger single motion-text datasets from Motion-X (Lin et al., 2024). This offers prior knowledge of how individual motions are described in language, enhancing the model's ability to align motions with corresponding textual descriptions.

**Instruction-tunning with INTER-MT$^2$ Data**  In this stage, we aim to enhance the model's ability to follow a wide range of instructions presented in a conversational format. We utilize INTER-MT$^2$ dataset combined with single-turn data from prior interactive motion datasets (Xu et al., 2024a; Liang et al., 2024). We follow instruction templates from MotionGPT (Jiang et al., 2023), to format the input and output for single-turn data. The multi-modal sequence $y$ consists of user instructions and corresponding responses, formatted as $y = (X_u^1, X_a^1, X_u^2, X_a^2, \cdots)$. The training objective remains the same as in the pre-training stage, with user instructions omitted in the loss function.

### 4.4 ADVANCED DOWNSTREAM INTERACTIVE MOTION TASKS

After training, our model can perform complex reasoning and generate interactive motions within multi-turn conversations. To verify this, we introduce two additional tasks requiring advanced capabilities: motion reasoning and editing. Motion reasoning involves predicting past or future events, or reasoning about current motions, based on prior conversational data. This task requires the model to understand the context of the conversation, interpret how the given motion fits within that context, and adjust its reasoning accordingly. In the motion editing task, we focus on altering a person's persona or shifting scenarios, such as emotions or relationship dynamics. This adds complexity, as changes to one person's behavior affect the other's motion. The model must edit the target motion while maintaining contextual coherence, requiring a deep understanding of social dynamics.

## 5 EXPERIMENTS

In our experiments, we evaluated VIM's ability to generate detailed motion-based chat responses, requiring complex reasoning about interactive motions, alongside traditional motion-related tasks. We focused on two main questions: first, whether the model can reason effectively about interactive motions, such as refining motions in editing tasks or generating contextually accurate narratives in motion reasoning. Second, we evaluated whether the training with INTER-MT$^2$ dataset improves the model's performance in text-to-motion, motion-to-text, and reaction generation tasks.

### 5.1 EVALUATION TASKS AND BASELINES

**Motion Reasoning**  Motion reasoning involves predicting past or future events or interpreting current motions using prior conversational context. We utilize powerful LLMs, i.e., GPT-4o (OpenAI, 2024) to assess the content alignment, naturalness, and logical coherence of the generated textual responses. Content alignment evaluates how accurately the text reflects the given motion data, logical coherence checks the consistency and reasoning accuracy of inferences made about past or future events, and naturalness evaluates the fluency of generated texts, with rating each metric on a 10-point scale. In addition, we utilized linguistic metrics like Rouge-L (Lin (2004)), METEOR (Banerjee & Lavie (2005)), and MAUVE (Pillutla et al. (2021)), to quantitatively assess the relevance, accuracy, and naturalness of the generated responses compared with labeled texts in INTER-MT$^2$ test dataset. We present the results on motion reasoning in §5.2.

**Motion Editing**  The goal of motion editing is to modify a reference motion based on user instructions. We conducted within-subject user studies to compare edited motion samples, with participants

Table 1: Evaluation on Motion Reasoning task with INTER-MT$^2$ test set. Coh., Align., and Nat. denote logical coherence, content alignment, and naturalness, respectively. **Bold** indicates best performance and underline denotes the second best performance.

| Methods | LLM-Assisted | | | Linguistic Metrics | | |
|---|---|---|---|---|---|---|
| | Coh. ↑ | Align. ↑ | Nat. ↑ | ROUGE-L | METEOR | MAUVE |
| *two-stage approach* | | | | | | |
| TM2T + LLaMA-3.1-8B | 3.852 | 3.050 | 6.348 | 0.158 | 0.226 | 0.009 |
| TM2T + GPT-4o | 4.266 | 3.455 | 6.790 | 0.162 | 0.227 | 0.019 |
| *unified approach* | | | | | | |
| MotionGPT$^*$ | 1.855 | 1.303 | 3.574 | 0.113 | 0.096 | 0.005 |
| MotionGPT$^*_I$ | 3.690 | 3.160 | 5.291 | 0.207 | 0.218 | 0.417 |
| VIM w/o INTER-MT$^2$ | 2.770 | 2.141 | 4.968 | 0.155 | 0.145 | 0.004 |
| **VIM (Ours)** | **5.252** | **4.511** | **6.981** | **0.239** | **0.260** | **0.794** |

rating them on three metrics: content similarity, instruction alignment, and motion quality, using a 5-point Likert scale, following Goel et al. (2024). Content similarity evaluates whether the edited motion preserves the original meaning of the source motion, while instruction alignment assesses how accurately the edited motion follows the given command. The study involved 30 participants, each evaluating five samples from a set of 30 randomly selected test samples. Participants evaluated four baselines and our method, viewing randomly shuffled motion outputs side by side and providing feedback on all metrics. We also employed data-driven metrics, such as Frechet Inception Distance (FID) and mean per joint position error (MPJPE) in meters, to evaluate the quality of the generated edited motion against the labeled motions in the INTER-MT$^2$ test set, following Goel et al. (2024). The results on motion editing is shown in §5.3.

**Traditional Motion Relevant Tasks** For standard motion-related tasks, we evaluated the proposed method in three traditional motion-relevant tasks in interactive motions: motion-to-text, text-to-motion, and reaction generation, in the union of the test set in InterHuman (Liang et al., 2024) and Inter-X (Xu et al., 2024a) datasets. To evaluate the text-motion matching score, we report the retrieval precision based on the feature space of retrieval models (Petrovich et al. (2023)). This evaluates the accuracy of matching between texts and motions using Top 3 retrieval accuracy with a fixed batch of 32. The quality of motion was measured by Frechet Inception Distance (FID), which measures the distance of feature distribution between motion data and generated motion. In addition, we measure the mean per joint position error (MPJPE) in meters to evaluate the accuracy of the reaction motion. The results on motion-related tasks are shown in §5.4.

**Baselines** Since our interactive multi-turn scenarios and tasks are novel, there is no exact comparison method. We consider and compare reasonable baselines that handle both motion and texts as input and output. We first employ a *two-stage approach* using off-the-shelf methods. For the motion reasoning task, we convert all motions into text descriptions using the state-of-the-art motion-to-text method TM2T (Guo et al., 2022). These textual descriptions are then used for text-based reasoning with large language models such as GPT-4o (OpenAI, 2024) and LLaMA-3.1-8B (Dubey et al., 2024). For the motion editing task, we combine the text converted by TM2T with the editing text command, and generate the edited motion using the off-the-shelf text-to-motion method InterGen (Liang et al., 2024). As baselines for the *unified approach*, we leverage MotionGPT (Jiang et al., 2023) framework with the following configurations: (1) MotionGPT$^*$: a modified MotionGPT fine-tuned on interaction data with instruction templates; (2) MotionGPT$^*_I$: MotionGPT$^*$ enhanced with INTER-MT$^2$ dataset; (3) VIM w/o INTER-MT$^2$: our method fine-tuned with instruction templates from MotionGPT, but without INTER-MT$^2$ data.

## 5.2 MOTION REASONING

In the motion reasoning task, conversations about two interactive motions are examined to assess the model's ability to deduce past or future events and comprehend the motivations driving the motions. The experimental results in Table 1 demonstrate that our unified model, VIM, significantly outperforms two-stage approaches across all LLM-assisted and linguistic metrics. Specifi-

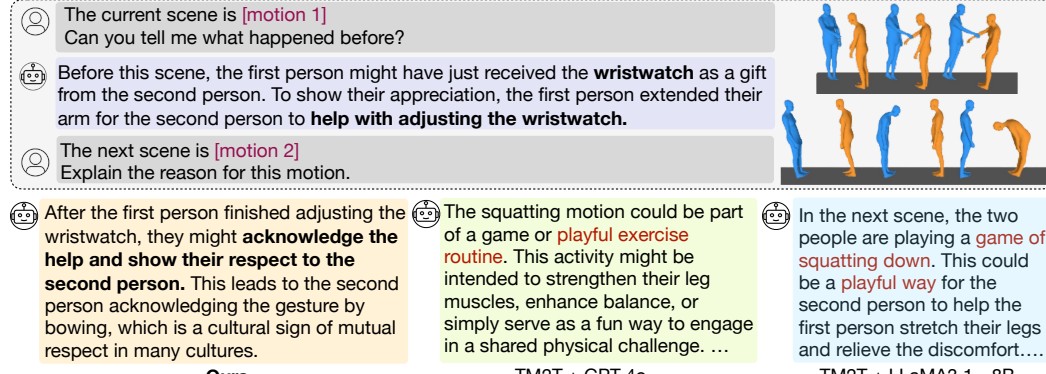

Figure 4: Generated samples for interactive motion reasoning task. This example shows how VIM explains behaviors and their motivations, demonstrating a deeper understanding of scenarios by incorporating context from prior interactions.

Table 2: Data-driven evaluation in motion editing in INTER-MT$^2$ test set.

| Methods | FID ↓ | MPJPE ↓ |
|---|---|---|
| *two-stage approach* | | |
| TM2T + InterGen | 0.110 | 0.811 |
| *unified approach* | | |
| MotionGPT$^*$ | 0.251 | 4.002 |
| MotionGPT$^*_I$ | 0.161 | 3.982 |
| VIM w/o INTER-MT$^2$ | 0.080 | 0.908 |
| **VIM (Ours)** | **0.064** | **0.758** |

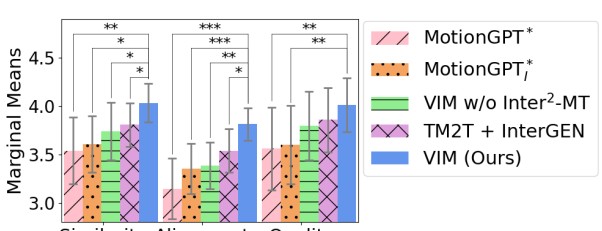

Figure 5: User subject study results for motion editing[1].

cally, VIM achieves improvements with performance increases exceeding 1.9 points in logical coherence, 1.1 points in content alignment, and nearly 0.2 points in naturalness compared to the best two-stage model. The baseline models trained solely on text-motion pair datasets, such as VIM w/o INTER-MT$^2$ and MotionGPT$^*$, show limited reasoning capabilities. Although MotionGPT$^*_I$, which incorporates INTER-MT$^2$ datasets, exhibits improved performance compared to baselines trained without INTER-MT$^2$, it still does not match the effectiveness of the two-stage approaches.

The improved performance of our unified model, VIM, over two-stage approaches, appears to result from two key factors: error accumulation and interpretation ambiguity. First, in two-stage models, errors can accumulate; if the motion captioning model generates incorrect motion captions, those mistakes carry over to the second stage, reducing content alignment and coherence. In contrast, VIM's unified architecture integrates motion encoding and reasoning in a single framework, minimizing error propagation. Second, interpreting motions is not always straightforward, with multiple ways to understand and describe the same motion. In two-stage models, mapping the motion to a single caption for the second stage can lead to more contextually accurate reasoning of the given scenarios or contexts. Our unified model, however, is built to recognize these varied interpretations and generate reasoning that is more contextually accurate. Figure 4 showcases it's ability to dynamically adjust interpretations and responses by incorporating context from previous conversations.

### 5.3 MOTION EDITING

We aim to validate the hypothesis that people will perceive the generated edited interactive motion from the proposed method to be more content-consistent, instruction-aligned, and better quality, through user subject studies. To analyze the results, we conducted a repeated-measures multivariate analysis of variance on the rated measures. We observed that methods significantly affect the user's

---

[1]We plotted the difference in a post hoc pairwise comparison of the proposed method only. We denote * as $0.01 < p < 0.05$, ** as $p < 0.01$, and *** as $p < 0.001$. The error bars represent 95% confidence intervals.

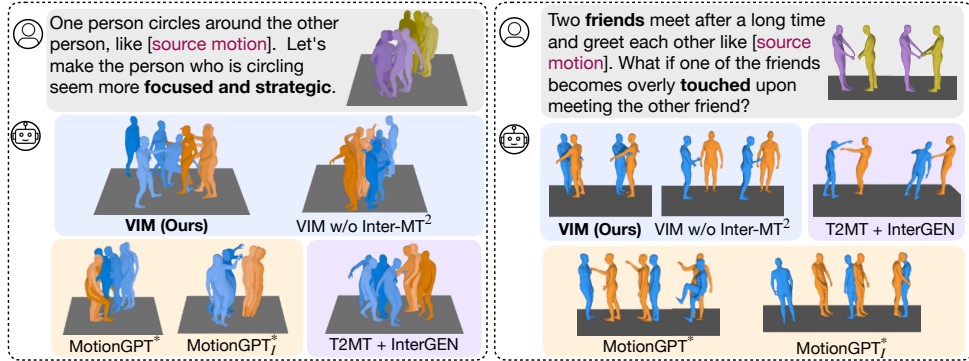

Figure 6: Generated samples for interactive motion editing. The proposed method excels in capturing nuances, outperforming alternatives in content similarity and instruction alignment.

Table 3: Comparisons for three motion-related tasks on Inter-X and InterHuman datasets. M2T denotes motion-to-text, T2M for text-to-motion, and Reaction Gen. for reaction generation.

| Methods | M2T | T2M | | Reaction Gen. | |
| --- | --- | --- | --- | --- | --- |
| | R Top3 ↑ | R Top3 ↑ | FID ↓ | MPJPE ↓ | FID ↓ |
| Real | 0.867 | 0.869 | 0.00 | - | 0.00 |
| MotionGPT$^*$ | 0.494 | 0.328 | 0.123 | 3.444 | 0.355 |
| MotionGPT$^*_I$ | 0.503 | 0.331 | 0.118 | 1.436 | 0.380 |
| VIM w/o INTER-MT$^2$ | 0.894 | 0.561 | 0.082 | 0.984 | 0.031 |
| **VIM (Ours)** | **0.901** | **0.568** | **0.059** | **0.691** | **0.019** |

perception of all dimensions; $F(4) = 4.591, p = 0.002, \eta^2 = 0.137$ for content similarity, $F(4) = 7.134, p = 0.000, \eta^2 = 0.197$ for instruction alignment, and $F(4) = 4.781, p = 0.001, \eta^2 = 0.142$ for motion quality, with all $\alpha = 0.05$. The estimated marginal mean of the rated score is reported in Figure 5. The results show that the proposed method had better instruction alignment, quality, and content consistency across other baselines with significant differences.

During post hoc pairwise comparisons, we identified a significant difference, with our proposed method outperforming the two-stage model (TM2T (Guo et al., 2022) with InterGEN (Liang et al., 2024)) in content similarity ($p = 0.017$) and instruction alignment ($p = 0.010$). The two-stage model showed lower content similarity due to motion-to-text conversion errors, leading to unintended motions, whereas our unified framework avoids such error accumulation. Additionally, the two-stage model struggled with instruction alignment because InterGEN was trained to generate motions from captions, limiting its ability to adapt to varying personas or contexts. In contrast, our method, trained on diverse instructions and tasks, demonstrated superior reasoning and adaptability, resulting in more accurate motion generation based on instructions and source motions.

In post hoc pairwise comparisons with MotionGPT$^*_I$, we observed significant differences, with our proposed method performing better in content similarity ($p = 0.005$), instruction alignment ($p < 0.0005$), and motion quality ($p = 0.009$). This suggests that the VQ-VAE-based tokenizer and conditional generation model negatively impacted performance. Additionally, compared to VIM w/o INTER-MT$^2$, there were significant differences in content similarity ($p = 0.010$) and instruction alignment ($p = 0.001$), indicating that without INTER-MT$^2$ data, the model struggles to control motion based on context and instructions. We also evaluated the proposed method using data-driven metrics, including FID and MPJPE, as shown in Table 2. The proposed method outperforms the baselines on both measures, which is consistent with the results from user studies. Figure 6 illustrates the generated edited motions based on the source motion and instruction.

## 5.4 TRADITIONAL MOTION RELATED TASKS

The results in Table 3 support our hypothesis that utilizing the INTER-MT$^2$ dataset enhances the model's performance in traditional motion tasks like motion-to-text (M2T), text-to-motion (T2M),

Table 4: Ablation Studies on motion tokenizer.

| Methods | Reasoning | | | Editing | | M2T | T2M | | Reaction Gen. | |
| | Coh. ↑ | Align. ↑ | Nat.↑ | MPJPE ↓ | FID ↓ | R Top3 ↑ | R Top3 ↑ | FID ↓ | MPJPE ↓ | FID ↓ |
|---|---|---|---|---|---|---|---|---|---|---|
| VIM-VQ | 5.004 | 4.256 | 6.915 | 0.892 | 0.128 | 0.861 | **0.601** | 0.101 | 1.109 | 0.055 |
| **VIM (Ours)** | **5.252** | **4.511** | **6.981** | **0.758** | **0.064** | **0.901** | 0.568 | **0.059** | **0.691** | **0.019** |

and reaction generation. The first row ("Real") shows retrieval accuracy, and FID scores from the dataset labels. Note that both VIM w/o INTER-MT$^2$ and MotionGPT$^*$ were trained on all of these tasks for fair comparison. Comparing the VIM w/o INTER-MT$^2$ to the version trained with INTER-MT$^2$ ("Ours"), we see improvements across all tasks. In M2T, Top-3 retrieval accuracy rose from 0.894 to 0.901. For T2M, Top-3 retrieval accuracy increased from 0.561 to 0.568, with FID dropping from 0.082 to 0.059, indicating better motion generation. In reaction generation, MPJPE dropped from 0.984 to 0.691, and FID from 0.031 to 0.019, confirming that multi-turn datasets improve motion comprehension and generation. Using the INTER-MT$^2$ dataset provides diverse, context-rich examples, helping the model learn more nuanced relationships between text and motion. Additionally, incorporating INTER-MT$^2$ in MotionGPT$^*$, denoted as MotionGPT$^*_I$, improved retrieval precision accuracy for M2T and T2M tasks, and joint position error in reaction generation.

## 5.5 ABLATION STUDIES ON MOTION TOKENIZER

We conducted ablation studies comparing the VQ-VAE-based model with our RQ-VAE-based approach, as shown in Table 4. The RQ-VAE-based motion tokenizer outperformed the VQ-VAE model in motion reasoning tasks, achieving higher scores in coherence, alignment, and naturalness. This improvement is attributed to reduced information loss, allowing our model to capture finer motion details while also enhancing its motion-to-text retrieval precision. For generation and editing tasks, the VQ-VAE model achieved slightly better text-to-motion retrieval accuracy but performed worse in FID and MPJPE across editing, reaction generation, and T2M tasks, indicating degraded motion quality and less precise motion details. In contrast, our approach reduced MPJPE by 0.055 for reaction generation, preserving joint dynamics and producing more realistic and natural motions. VQ-VAE's limitations are especially problematic for modeling interactive motions, where precise relative positioning is crucial, making its information loss and reconstruction quality more evident.

## 6 CONCLUSION AND DISCUSSIONS

**Conclusion** In this paper, we introduced VIM, a versatile motion-language model designed to model, understand, and reason about interactive motions. We outlined its architecture and provided detailed training strategies to create a unified framework integrating large language models with interactive motion modality. To enhance the model's reasoning capabilities and versatility, we presented a specialized dataset, INTER-MT$^2$, which incorporates a variety of reasoning tasks set within multi-turn conversations centered on interactive motions. Our experiments demonstrated VIM's ability to effectively follow instructions, edit motions, and reason about interactive motions.

**Limitations and Impact Statement** There are several limitations that warrant attention. First, the model's expressiveness remains limited when handling complex or previously unseen actions, indicating a need for further diverse motion source data in its ability to generalize across diverse motion scenarios. Second, the sequence length becomes excessively long as we flatten the residual motion tokens, which can impact efficiency and computational resources. Leveraging additional transformer models to predict the residual token can reduce this work. Lastly, our method faces challenges in personalization and interpretability, as motion is inherently ambiguous and users may interpret the same motion in different ways. Addressing this issue will require incorporating more tailored approaches that adapt to individual user preferences and expectations through further human-in-the-loop feedback and refinement processes. In terms of broader impact, VIM opens up new possibilities for interactive motion modeling and understanding in AI, potentially benefiting fields like robotics, virtual environments, and human-computer interaction. However, careful consideration of ethical concerns, such as misinterpretation of motions or unintended behavioral biases, is crucial as the model evolves.

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
