# OpenReview forum: "Versatile Motion-Language Models for Multi-turn Interactive Agents"
_ICLR.cc/2025/Conference — ICLR 2025 Conference Withdrawn Submission_

### Official Review · Reviewer_ztzV · 2024-10-18

**Soundness:** 1
**Presentation:** 1
**Contribution:** 1
**Rating:** 3
**Confidence:** 5

**Summary:**

This paper proposes an LLM-based motion generation and understanding method, which supports multi-round conversation and reasoning. This paper also proposes a GPT-4o-generated dataset to support the model training for this task. The authors also provide some experiments to support the claim.

**Strengths:**

Introducing LLM into the animation is a good topic.

Limitations and impacts are discussed.

**Weaknesses:**

- One of the key concerns is the dataset quality. 1) Are all inputs of GPT-4o in Fig. 2a are text-only? If so, how do authors generate the "user input" shown on the left of Fig. 2a? 2) As shown in Fig. 2b, how can intergen generate two motions with the single prompt (see the generated result of caption 1)? 3) The GPT-4o always exists some hallucinations, such as "sorry/ I don't know", which result in the failure cases in your dataset. Has the data been checked by researchers one by one? Besides, the motion data here is also generated, which reduces the data quality for training.

- The technical soundness.
    - The authors use the RVQ to compress motions. As stated in the main text, the motion tokens are contacted with texts into the LLM jointly. I have some concerns about this. If $\mathbf{z}$ is one compressed token by the encoder, and it is quantized by $\{{\bf z}_1, {\bf z}_2\}$, aka. $\mathbf{z} = {\bf z}_1+ {\bf z}_2 + \delta$, where $\delta$ is the quantization error. According to the statement in the paper, the method feeds ${\bf z}_1, {\bf z}_2$ directly into the LLM. However, the method can feed ${\bf z}_1+ {\bf z}_2$ into LLM directly, which is more token efficient and equal to the method proposed by authors.
    - A lot of research verifies that the adaption between vision tokens and language tokens is needed. For example, MotionLLM and LLaVA series. Therefore, the technical design here is surprising. An ablation is needed.

- No video demo provided for animation is not acceptable.

- The multi-round interaction is one of the main claims of the paper, which is not thoroughly verified by experiments. The reasoning cases in Figure are limited. The results of more interaction turns should be provided. Besides, As shown in the Fig. 4, how does your method insert the motion into the text sequence? How do users determine the position to insert? The paper lacks a clear statement for user interaction.

- It is not clear about the detailed evaluation protocol in Table 1. What is the test set (size)? As shown in the appendix, the score ranges from 0 to 10. However, the scores shown in Table 1 are quite poor. What is the key contribution of the evaluation protocol claimed in L101-102?

- The authors **have fundamental errors** in discussing related works. In L125-126, why does Chen et al. (2024a) include speech and music? Besides, this is the most related work about motion-LLM that authors should discuss the difference.

- The writing issue. In L142-143, "Xu et al. (2024b)" should be "(Xu et al. 2024b)".

This work lacks unique contribution and a lot of details are not clear or well-discussed. Therefore, I vote to reject clearly.

**Questions:**

This question is very important. The motion tokens are not in the LLM vocabulary and the dimension differences are not clear. How do authors resolve this issue? Have authors expand the vocabulary size? Besides, how does the method identify which token is motion or text? As shown in Fig. 3, is the output sequence of the method motion-only or text-only, without text and motion jointly?

The dataset does not include motion editing. How can your method perform the editing? This is quite confusing.

---

### Official Review · Reviewer_sFZK · 2024-11-01

**Soundness:** 1
**Presentation:** 3
**Contribution:** 3
**Rating:** 5
**Confidence:** 5

**Summary:**

This paper is addressing a novel task, conversation-based human interaction synthesis, understanding, and editing. To approach this task, the authors first build a synthetic human interaction datasets by employing GPT-4o to produce instructions and captions, and by using interGEN to synthesize human interactions. For the interaction-language model, this paper adopted a motiongpt-like framework, where the interactions are first encoded into discrete (residual) motion tokens, and then combined with text tokens. Then, the motion vocabulary is appended to the text vocabulary. A pre-trained LLM model is finetuned with motion tokens. After all, the proposed framework is versatile, being capable of interaction understanding, editing, and generation.

**Strengths:**

**1. Challenging, yet Novel and interesting task**.  Some settings of this paper is similar to MotionGPT, while the current work is targeting two-person interactions. As known, modeling interactions are way more complicated than single-person motion synthesis. The current work makes a novel and important trial towards this direction.
**2. Dataset Inter-MT^2 is valuable, the way for creating the dataset is insightful**. Since there is no proper interaction dataset for current context, the authors propose to use instruction templates and LLM to generate captions and conversations, and use interGEN to generate corresponding interactions data. Though this would highly rely on the performance of interGEN, it provides new insight on how to create motion and interaction that are typically hard to obtain.
**3. Leveraging LLM for two-person interactions generation is innovative**. This paper achieves multiple interaction related tasks on the top of the existing large language model. This way of modeling interaction is novel and insightful.
**4. One framework for multiple interaction related tasks**. This one model can handle interaction editing, interaction understanding, interaction synthesis all in one. The versatility of this model is appreciated.
**5. Presentation is clear. This paper is well-organized**. The presentation is easy to follow.
**6. Quatitative evaluation results looks promising**.

**Weaknesses:**

**1. Lack of qualitative results**. The biggest concern is the lack of demo video in the submission. I didn't see either video or the link to videos. I could see many numbers in tables, visualization in figures. Nevertheless, without video demonstration, it's hard to evaluate the real performance, especially for interaction synthesis.
**2. The quality of dataset is unknown**. Since the inter-MT^2 is based the generation of interGEN, it is hard to guarantee the overall quality of the interaction dataset. Whether it's faithful to the text, or if the interaction is natural.  Some examples and stats about this dataset will be appreciated.
**3. The effectiveness of modeling motion token using LLM is unknown**. It is questionable if it's really effective to merge the motion tokens and text tokens together, given that motion token and text token are substantially different modalities.
**4. Some typos**. In the paper, "TM2T" is written as "T2MT" in several places, e.g., Figure 6.
**5. Baseline implementations are missing**. TM2T and MotionGPT are both designed for single person motion generation. It is not clear how they are adopted in current context. For example, have you changed the model architecture? Some details regarding the implementation are appreciated.

**Questions:**

Please refer to the weakness section. My main concern is the lack of video demonstration. However, I don't think it is fair for other authors if the videos are submitted during rebuttal period.

---

### Official Review · Reviewer_eXv2 · 2024-11-04

**Soundness:** 2
**Presentation:** 3
**Contribution:** 2
**Rating:** 5
**Confidence:** 4

**Summary:**

In this paper, the authors present the Versatile Interactive Motion (VIM) language model, developed to manage interactive human motions in multi-turn conversational contexts. By merging linguistic and motion-based modalities, VIM demonstrates the ability to comprehend, generate, and adjust complex interactive human motions. To address the scarcity of interactive motion data, the authors introduce INTER-MT2, a synthetic dataset featuring multi-turn interactions with dynamic motion sequences. The model’s capabilities are illustrated through tasks such as motion reasoning, editing, captioning, and related tasks.

**Strengths:**

1. The motivation to explore multi-turn conversations in interactive motion generation is convincing.
2. Extensive experiments and ablation studies are conducted, demonstrating the superior performance of the proposed model.
3. The paper introduces a multi-turn conversational motion dataset, which enables the effective training of the language model for interactive motion generation within conversations.

**Weaknesses:**

1. While the unified motion-language generative model is a notable concept, the method is overly conventional within the multimodal large language model (MLLM) field, lacking technical contributions specifically designed for interactive human motion.

2. The experiments on the proposed dataset in motion-related tasks, as shown in Table 3, are insufficient. The paper falls short of demonstrating the impact of synthetic conversational data on motion-related tasks, to validate the effect of the proposed dataset.

3. In Table 4, the ablation studies on the motion tokenizer require comparisons with state-of-the-art RVQ-VAE models, such as MoMask [1]. Given that the proposed approach also utilizes RVQ-VAE, such comparisons would provide a more comprehensive understanding of the model’s relative effectiveness.

4. The paper provides only static visualizations of the model’s results, which limits the clarity of the motion generation process. Including dynamic visualizations would offer a more intuitive and detailed depiction of the generated motions.

[1] MoMask: Generative Masked Modeling of 3D Human Motions

**Questions:**

1. I am curious about how the training and inference efficiency of the proposed method compared to existing models. Could you provide more details about this?
2. It would be interesting to see how the performance of the proposed method varies with increasing conversational turns and extended motion sequences.

---

### Official Review · Reviewer_maA9 · 2024-11-04

**Soundness:** 3
**Presentation:** 3
**Contribution:** 3
**Rating:** 5
**Confidence:** 3

**Summary:**

The authors introduce a synthetic dataset, **INTER-MT$^2$**, for multi-turn interactive motion generation and understanding. Moreover, this paper proposes a Versatile Interactive Motion language model, **VIM**, to integrate both text and motion modalities for unifying the motion understanding and generation task.

**Strengths:**

This paper uses the residual discrete tokens to represent human motion and designs suitable templates for effective learning of text and motion. The unified model facilitates many downstream tasks, such as motion reasoning, editing, etc. Experiment results show the method is robust to handling complex interactive motion synthesis and understanding.

**Weaknesses:**

The architecture of VIM is pretty similar to LLaVA, so nothing new here. I’m more interested in the multi-turn conversation dataset. Could you show more examples of the videos generated by the model? Also, does the text generated by GPT-4o not need any data cleaning? Most of the motions in this dataset are generated by the text2motion diffusion model, so the quality obviously can’t be guaranteed. As for InterGEN in Figure 1b, can it generalize well enough to the text generated by GPT-4o? I have my doubts.

**Questions:**

Missing cites:
- EMDM: Efficient Motion Diffusion Model for Fast and High-Quality Motion Generation
- MotionLCM: Real-time Controllable Motion Generation via Latent Consistency Model
- StableMoFusion: Towards Robust and Efficient Diffusion-based Motion Generation Framework

---

### Note · Authors · 2024-11-14

I have read and agree with the venue's withdrawal policy on behalf of myself and my co-authors.